# Neural Mechanisms of Cancer Cachexia

**DOI:** 10.3390/cancers13163990

**Published:** 2021-08-07

**Authors:** Brennan Olson, Parham Diba, Tetiana Korzun, Daniel L. Marks

**Affiliations:** 1Medical Scientist Training Program, Oregon Health & Science University, 3181 SW Sam Jackson Park Rd, Portland, OR 97239, USA; olsobr@ohsu.edu (B.O.); diba@ohsu.edu (P.D.); korzun@ohsu.edu (T.K.); 2Papé Family Pediatric Research Institute, Oregon Health & Science University, 3181 SW Sam Jackson Park Rd, Portland, OR 97239, USA; 3Brenden-Colson Center for Pancreatic Care, Oregon Health & Science University, 3181 SW Sam Jackson Park Rd, Portland, OR 97239, USA

**Keywords:** cancer, cachexia, cytokines, neuroinflammation, autonomic nervous system, neuroendocrinology, GDF15, LCN2, INSL3

## Abstract

**Simple Summary:**

Cancer cachexia is a devastating wasting syndrome that occurs in many illnesses, with signs and symptoms including anorexia, weight loss, cognitive impairment and fatigue. The brain is capable of exerting overarching homeostatic control of whole-body metabolism and is increasingly being recognized as an important mediator of cancer cachexia. Given the increased recognition and discovery of neural mechanisms of cancer cachexia, we sought to provide an in-depth review and update of mechanisms by which the brain initiates and propagates cancer cachexia programs. Furthermore, recent work has identified new molecular mediators of cachexia that exert their effects through their direct interaction with the brain. Therefore, this review will summarize neural mechanisms of cachexia and discuss recently identified neural mediators of cancer cachexia.

**Abstract:**

Nearly half of cancer patients suffer from cachexia, a metabolic syndrome characterized by progressive atrophy of fat and lean body mass. This state of excess catabolism decreases quality of life, ability to tolerate treatment and eventual survival, yet no effective therapies exist. Although the central nervous system (CNS) orchestrates several manifestations of cachexia, the precise mechanisms of neural dysfunction during cachexia are still being unveiled. Herein, we summarize the cellular and molecular mechanisms of CNS dysfunction during cancer cachexia with a focus on inflammatory, autonomic and neuroendocrine processes and end with a discussion of recently identified CNS mediators of cachexia, including GDF15, LCN2 and INSL3.

## 1. Introduction

Although cachexia is a multi-organ syndrome involving complex inter-organ interactions during its progression, the central nervous system (CNS) is uniquely equipped in exerting overarching homeostatic control of peripheral tissues through its direct control of illness behaviors (such as anorexia and fatigue), efferent engagement of the autonomic nervous system and regulation of neuroendocrine axes [1,2,3]. Furthermore, several decades of research demonstrate that the CNS is both a receiver and amplifier of peripheral inflammatory signals and that this amplification of peripheral signals is responsible, in part, for regulating several metabolic and behavioral manifestations of cachexia [4,5]. Research dedicated to unveiling CNS mechanisms of cachexia has historically focused on the hypothalamus, a central coordinator of several homeostatic processes that are known to be awry during cachexia, including appetite, sleep, activity level, wakefulness and macronutrient distribution, to name a few. However, recent work in this field has implicated additional CNS structures involved in cachexia symptoms, including the brainstem and parabrachial nucleus [2,6,7]. These advances in the field demonstrate that not only is our understanding of the hypothalamic mechanisms of cachexia incomplete at this time, but also that our understanding of how other brain structures and their CNS circuitry influence cachexia is in its infancy. Nevertheless, given the clear role of the CNS in the development of the signs and symptoms of cachexia, combined with recent advances in CNS-based mechanisms of cachexia, research devoted to unveiling aberrant CNS pathways during cachexia represents a promising approach in identifying therapeutic targets for this metabolic syndrome. This review highlights studies investigating how the brain mediates cancer cachexia, with a focus on foundational work in inflammatory, autonomic nervous system and neuroendocrine mechanisms. We end this review with a discussion of recently identified central mediators of cancer cachexia, including Growth and Differentiation Factor 15, Lipocalin 2 and Insulin-like 3 peptide, and their intersections with the aforementioned CNS pathways.

## 2. Central Inflammation: Lessons from IL-1β

Cancer, as well as the cytotoxic chemotherapy utilized to treat the cancer, is often accompanied by prolonged systemic inflammation. As such, it is generally accepted that inflammation is a key component in the development and progression of cancer cachexia [8,9,10]. Indeed, increased circulating levels of inflammatory cytokines (e.g., interleukin-6 (IL-6) [11], tumor necrosis factor alpha (TNF-α) [12] and C-reactive protein [11]) and innate immune cells (e.g., neutrophils [13]) are all associated with cachexia in humans. These systemic inflammatory mediators are derived from several host tissue sources, including the liver, fat, skeletal muscle and bone marrow. Several systemic mediators are also produced by developing tumor, including nearby stromal cells in response to the tumor, infiltrating immune cells and neoplastic cells themselves [14,15,16,17,18] (Table 1; newly identified mediators are discussed in the final section of this review). Although beyond the scope of this review, it is worth noting the emerging link between metastatic disease and cancer cachexia, in which metastatic invasion into tissues (including the brain) may result in local inflammatory niches conducive to cachexia-associated wasting, although the impact of individual metastases on cachexia development remains elusive [19,20,21]. In addition to peripherally derived inflammatory molecules, a recent report describes a distinct neuroimmune axis in driving cachexia by which peripheral myeloid cells directly invade the CNS [7]. Once produced in the circulation, these molecular and cellular inflammatory mediators are able to either interface with the brain by directly crossing the blood–brain barrier (BBB) [22] or through direct blood-borne sampling by circumventricular organs (CVOs).

Seminal work describing the cachexia-inducing potential of systemic inflammatory mediators originated from simple experiments in which cytokines were administered directly to the brain of rodents. Specifically, intracerebroventricular (ICV) injection of pathophysiological levels of interleukin 1 beta (IL-1β) or TNF-α results in anorexia, weight loss, increased energy expenditure and accelerated catabolism of fat and lean mass [42,43]. Furthermore, the central administration of these cytokines also results in a paracrine loop, through which their administration into the brain leads to an increase in endogenous production, both maintaining and propagating a local inflammatory milieu in the CNS [44,45,46]. These foundational experiments demonstrate that when inflammatory cytokines produced during cachexia interface with the CNS, (1) illness behaviors consistent with cachexia are individually produced and (2) the brain not only receives these inflammatory signals, but interprets and amplifies these signals in proximity to CNS nuclei integral in the regulation of energy homeostasis (Figure 1).

Of the inflammatory cytokines produced peripherally and centrally during cachexia, possibly no other is more studied than IL-1β. Indeed, several studies show that IL-1β is the major cytokine induced in the mediobasal hypothalamus as a result of peripheral tumor development [3,47,48]. These and many other reports formed the precedent for the study on IL-1β in the development of illness behaviors and cachexia over the past two decades. It was recently demonstrated that brain-endothelial expression of the interleukin-1 receptor (IL-1R) enhances leukocyte recruitment, mediates sickness behavior and impairs neurogenesis [49]. Brain endothelial cells respond to IL-1β in a myeloid differentiation primary response protein (MyD88)-dependent manner, which amplifies and propagates inflammatory signals to glial cells [50]. The role of IL-1β in cachexia is further demonstrated by cachexia studies in which MyD88, which is the universal adaptor protein to all Toll-like receptors (TLRs; except TLR3) and the interleukin 1 receptor family, is implicated in the pathogenesis of cancer cachexia. In two separate reports utilizing different models of cancer cachexia, *MyD88* deletion attenuated several measures of cachexia, including anorexia, muscle catabolism, fat loss, hypothalamic inflammation and fatigue [51,52]. In contrast, a recent report demonstrated that genetic deletion *Il-1β* failed to improve fatigue symptoms in several different rodent models of cancer, suggesting that, while IL-1β signaling may initiate or drive some cachexia symptoms, it is unlikely to be an all-encompassing therapeutic target [53]. Given the robust induction of several inflammatory cytokines in the context of cancer cachexia, it is plausible that a combinatorial blockade of central-acting inflammatory mediators is requisite in mitigating cachexia. While we utilized IL-1β as a conceptual framework for CNS bioamplification of peripheral signals during cachexia, there are undoubtedly other inflammatory cytokines that undergo similar amplification events by the brain, including TNF-α, IL-6 and leukemia inhibitory factor (LIF)—mechanistic review of these cytokines in the brain can be found in recent reviews [2,5].

In addition to cytokines being received and amplified in the CNS during cachexia, a recent report from our lab demonstrates a clear role for myeloid cell invasion in the CNS in driving pancreatic cancer cachexia symptoms, including anorexia and lean mass catabolism, that is predominantly driven by the CCR2–CCL2 axis [7]. Prior reports demonstrate a clear role for CNS-infiltrating immune cells in both health and disease, having either beneficial or detrimental effects, depending on the underlying pathology [54]. In this study, the majority of the CNS-invading immune cells were neutrophils, which accumulated at a unique CNS structure called the velum interpositum (VI). Interestingly, a large percentage of neutrophils in this region expressed CCR2, which is typically considered a monocyte chemotaxis receptor. CCR2 deletion attenuated cachexia and prevented neutrophils from infiltrating the VI during pancreatic cancer cachexia. Furthermore, these CNS-invading neutrophils expressed a transcriptome that is dissimilar from that of neutrophils invading peripheral tissues, implicating a distinct neutrophil population infiltrates the brain during cancer cachexia. Although future investigation is needed, this study demonstrates that the inflammatory mediators during cachexia may extend beyond canonical cytokines to CNS-invading immune cells, by which myeloid cells (namely neutrophils) enter the CNS through a unique meningeal portal to incite inflammation in CNS structures important in energy homeostasis.

Although blockade of cytokine signaling in rodent models of cancer cachexia has yielded promising results, concomitant clinical trials have broadly failed to attenuate cancer cachexia in humans [55]. Thus, while systemic inflammatory mediators may be important in the initiation of illness behaviors and cachexia, current data suggest that they are insufficient in sustaining cachexia [56,57,58,59]. These results demonstrate the complexity of cancer cachexia and, although they do not preclude anti-inflammatory therapies in its treatment, they suggest they may have limited utility as a monotherapy.

## 3. Sympathetic Nervous System Engagement

In response to stress, the CNS coordinates an evolutionarily conserved fight-or-flight response, a metabolically costly physiologic program that rapidly liberates energy stores for muscle use. This program is directed within the CNS, where neurons in the paraventricular nucleus of the hypothalamus (PVH) integrate environmental and neuronal input and project to noradrenergic centers in the brainstem and preganglionic neurons in the spinal cord [60]. Furthermore, extensive remodeling of the sympathetic nervous system (SNS) during chronic inflammatory stress greatly modifies end-organ responses to SNS inputs [47,61,62]. Acting through sympathetic ganglia throughout the body, the SNS regulates end organs via the release of norepinephrine at synapses and release of epinephrine from the adrenal medulla into the circulation. These neurotransmitters elevate metabolic rate by increasing heart rate and cardiac contractility, engaging lipolysis in white adipose tissue (WAT) and inducing non-exercise thermogenesis in brown adipose tissue (BAT) by uncoupling electron transport from ATP generation in the mitochondria [63]. Indeed, independent studies demonstrated that sympathetic tone is elevated in patients and mice with cachexia and a sustained elevation in basal metabolic rate is repeatedly described as a critical energy-expenditure mechanism during cancer cachexia. Despite the clear involvement of SNS activation in the progression of cancer cachexia, the precise mechanisms of SNS engagement in cachexia remain incompletely understood [64,65]. Here we will summarize known and potential mechanisms by which the CNS orchestrates SNS activation, tissue remodeling and energy wasting in cachexia (Figure 2).

Increased BAT thermogenesis in cachectic mice was first reported 40 years ago and has since been reported in multiple murine models of cachexia [64,66,67]. In addition to increased BAT thermogenesis, excess energy expenditure in cachexia can be mediated by “browning” of WAT, in which WAT takes on some of the molecular characteristics and thermogenic capability of BAT—a process mediated by SNS inputs [68,69]. Indeed, it was recently demonstrated that the β3-adrenergic receptor blockade—the receptor responsible, in part, for establishing sympathetic tone in tissues—ameliorated adipose wasting, browning and cachexia-associated weight loss [61]. While this study implicates the SNS in remodeling adipose tissue, resulting in the upregulation of energetically expensive thermogenesis programs, it remains unclear which CNS pathways are responsible for this increased SNS tone during cachexia development. Recently, a brain circuit was identified that potently regulates metabolism in WAT and BAT [68]. This circuit is initiated in the arcuate nucleus (Arc) of the MBH via the interaction of pro-opiomelanocortin (POMC) and Agouti-related peptide (AgRP) neurons with the adipokine leptin, then relayed via paraventricular hypothalamic (PVH) BDNF neurons that ultimately regulate SNS tone in peripheral adipose tissues. Interestingly, these authors demonstrate remarkable plasticity in SNS innervation of adipose depots and this plasticity is dependent on chronic (rather than acute) signaling in the Arc. Additionally, a recent report by Kim and colleagues shows a precise and rapid activation of adipose tissue lipolysis by the brain–fat axis in the context of bacterial infection and that this axis is entirely dependent on hypothalamic Arc TNF receptor activation and sympathetic nerve outflow to fat [70]. Since TNF, also known as cachectin, is strongly implicated as a mediator of CNS inflammation during cachexia, it is plausible that this described brain–fat axis mediates some of the adipose tissue remodeling and wasting observed during cancer cachexia. Even though the authors described this phenomenon in the context of adaptive immune response and did not investigate browning signatures in WAT, it seems plausible that the chronic inflammatory state induced by cachexia could induce sustained activation of this brain–fat axis through hypothalamic TNF signaling, resulting in the prolonged lipolysis and progressive fat wasting typical of cachexia.

Further evidence of SNS activation in cachexia is derived from studies of HR variability (HRV) in cachectic patients. HRV is measured as variance in time between successive beats and is a measure of autonomic tone. Imbalances in autonomic tone (most commonly elevated sympathetic input) decrease HRV and are associated with increased mortality. Cachectic cancer patients were found to have substantially decreased HRV, indicating elevated SNS tone [71,72]. A recent report by Luan and colleagues demonstrate a brain–fat–heart axis in the maintenance of stroke volume and overall cardiac output in the context of acute inflammation after lipopolysaccharide challenge [73]. The authors identify Growth and Differentiation Factor 15 (GDF15) as the critical mediator of this axis. Specifically, GDF15 is produced in the liver during acute inflammation and secreted into circulation where it then binds to its receptor in the area postrema of the brainstem, resulting in SNS outflow back to hepatic tissue and subsequent mobilization of lipids. The authors demonstrate that this lipid mobilization is critical in the maintenance of the cardiovascular function during acute inflammation and intricately demonstrate that hepatic β-adrenergic signaling is central to this process [73]. Since GDF15 is known to be upregulated in numerous rodent cachexia models and humans with cancer, it is likely that GDF15-driven SNS engagement mediates wasting during cancer cachexia as a recent report suggests [74,75,76]. Further discussion of the therapeutic promise of GDF15 will be provided in the final section of this review.

During acute stress, transient activation of the stress response elicits metabolic and behavioral adaptations that are beneficial to the organism over the short term. In chronic disease states, prolonged activation of the stress response is maladaptive and leads to loss of physiologic reserve and cachexia. Given the SNS plays an essential role in the acute stress response and recent work implicates SNS engagement in the pathogenesis of cancer-associated wasting, the identification of molecular mediators and pathways in which the SNS is chronically activated represents a promising approach to treating this debilitating wasting syndrome. Further research in this space is needed.

## 4. Neuroendocrine Modulation

The CNS is also a central regulator of endocrine organ function through the release of several hypothalamic-pituitary hormones, including the corticotropin-releasing hormone (CRH), thyroid releasing hormone (TRH) and gonadotropin-releasing hormone (GnRH), to name a few. These hypothalamic neuroendocrine neurons all send projections to the fenestrated capillaries of the median eminence, where they secrete hormones into the portal system, ultimately acting on the pituitary gland to amplify additional hormone secretion into peripheral circulation. While the neuroendocrine regulation of behavioral aspects of cachexia is well established (for reviews, see [77,78]), the cachexia field is still unveiling mechanisms by which aberrant endocrine function leads to the direct catabolism of lean and fat tissues. Here we will review these recent reports and mechanisms by which CNS control of endocrine response mediates cachexia-associated wasting (Figure 3).

The hypothalamic–pituitary–adrenal (HPA) axis is activated by a wide array of stressors, including fear, fasting or undernutrition, acute illness and injury. Since many of these stressors exist in the context of cachexia, it stands to reason that the HPA axis is engaged during cachexia. Indeed, several reports show an increase in the glucocorticoid (GC) corticosterone—the rodent analog of human cortisol—in mice with cancer cachexia [47,79]. GCs are released as the systemic effector molecules of the neuroendocrine arm of the CNS stress response beginning with activation of CRH neurons in the PVH and leading to the release of cortisol (in humans) or corticosterone (in mice) from the adrenal cortex. GCs are a primary driver of skeletal muscle catabolism due to acute inflammation and denervation injury and exogenous GCs are sufficient to drive muscle catabolism in both mouse and human [3,80]. Our laboratory showed that GC ablation, antagonism or deletion of the GC receptor (GR) in the muscle all prevented inflammation-, lung cancer-, or chemotherapy-associated muscle wasting [3,80,81]. Thus, GCs act as a required permissive factor to allow pathological muscle catabolism. Mechanistically, GCs induce muscle wasting by both promoting catabolism by transactivation of *Foxo1* and *Trim63* and by blocking the activity of the mammalian target of rapamycin (mTOR), a regulator of muscle anabolism [82,83]. These reports strongly implicate GC signaling in driving cancer-associated muscle wasting and demonstrate that the CNS maintains endocrine control of, at a minimum, skeletal muscle wasting.

In addition to CNS control of the HPA axis, induction of GCs and subsequent muscle atrophy during cancer cachexia, recent reports also suggest dysfunction of the hypothalamic–pituitary–gonadal (HPG) axis in cancer patients [84,85]. These reports detail a significant decrease in testosterone levels in male cancer patients, both prior to and after the initiation of cancer treatment, estimating that 40–90% of cancer patients suffer from low testosterone levels [86,87]. In a murine model of pancreatic cancer cachexia, male tumor-engrafted mice displayed a loss of over 97% of circulating free testosterone, relative to control mice [47]. Given the known anabolic properties of testosterone, this hypogonadal state likely contributes to the propagation of cachexia symptoms, including fatigue, weight loss and muscle catabolism. Indeed, Skipworth and colleagues recently demonstrated that hypogonadal cancer patients experienced significantly greater weight loss than eugonadal patients [88], while Dev and colleagues demonstrated low testosterone levels were associated with increased systemic inflammation, weight loss and decreased overall survival [89]. Despite the field’s acknowledgement of hypogonadism in cancer cachexia, little mechanistic advancement has been made into how cancer and its therapeutics disrupt the HPG axis. A recent review by Burney and Garcia postulated several potential mechanisms by which the HPG axis could be altered during cancer cachexia and included inflammation and hypothalamic response to leptin as potential modulators of the HPG [84]. However, future mechanistic research is needed to determine how cancer enacts neuroendocrine change through the HPG axis.

The hypothalamus is the master regulator of the endocrine system through its ability to detect peripheral signals and produce a robust systemic hormonal response through several axes, including the HPA and HPG ones. Since endocrine dysfunction is now recognized in cachexia as a potent regulator of peripheral tissue catabolism, it is plausible that other endocrine aberrations contribute to cachexia-associated wasting.

## 5. Recently Identified CNS Mediators of Cachexia: GDF15, LCN2 and INSL3

With the CNS being integral in mediating several facets of cancer cachexia, including illness behaviors and homeostatic control of muscle and adipose tissue metabolism, it logically follows that peripherally derived factors produced from either the tumor or host tissues interface with the brain to drive some of these signs and symptoms of cachexia. Indeed, just the past 2 years of research identified GDF15, Lipocalin 2 (LCN2) and insulin-like 3 peptide (INSL3) as novel peripherally derived factors that activate CNS circuitry to drive cachexia symptoms (Figure 4). Here, we will briefly discuss these novel mediators of cachexia through their actions in the CNS and future directions for their study.

### 5.1. GDF15

GDF15 is a circulating protein implicated in feeding behaviors and metabolism through its actions on the orphan receptor GDNF family receptor α-like (GFRAL) in the hindbrain [90,91]. Since the discovery of GFRAL as the receptor for GDF15 in 2017, numerous reports have furthered the mechanistic basis for the GDF15–GFRAL axis in food intake and metabolism. Utilizing shrews—a rodent species capable of vomiting—Borner and colleagues eloquently demonstrated that the anorexigenic effects of GDF15 are mediated through nausea and emesis after its binding to GFRAL in the area postrema (also known as the vomiting center of the brain) [92]. This study provided the basis for the anorexia-mediating effects of the molecule, but a report by Luan and colleagues, in 2019, also demonstrated a nutrition-independent role of the GDF15–GFRAL axis [73]. Specifically, the authors demonstrate that GDF15 binding to GFRAL in the brainstem results in autonomic outflow to the liver and subsequent mobilization of lipids, a triglyceride metabolism axis that is cardioprotective in the context of acute inflammation [73]. Given the putative roles of GDF15 in regulating appetite and fat metabolism, two important facets of cancer cachexia, it was hypothesized that inhibition of the GDF15–GFRAL axis could modulate the trajectory of cancer cachexia.

A report by Suriben and colleagues, in 2020, demonstrated that inhibition of GDF15 activation of GFRAL in the brainstem, using a monoclonal antibody (3P10) that inhibited RET recruitment to the GDF15–GFRAL complex, significantly curtailed excessive lipid oxidation in several murine cancer cachexia models [76]. Furthermore, the authors demonstrated that pair feeding 3P10-treated tumor-bearing mice to IgG-control tumor-bearing mice still led to a significant improvement in body mass, revealing a food-intake independent mechanism of GDF15 during cancer cachexia [76]. The authors concluded that GDF15 fat-mobilizing effects are mediated by SNS activation of adipose triglyceride lipase (ATGL) and hormone-sensitive lipase (HSL) in white adipose tissue [76]. This work demonstrates that, in combination with GDF15 anorectic effects during cancer cachexia [75], inhibition of brainstem activation of GFRAL by GDF15 mitigated cancer cachexia-associated wasting through dampening activation of the sympathetic nervous system. Since GDF15 appears to influence multiple aspects of CNS-mediated cachexia, including appetite, autonomic outflow and subsequent fat mass wasting, blocking GDF15 signaling represents a promising anti-cachexia strategy.

### 5.2. LCN2

Lipocalin 2 (LCN2), also known as neutrophil gelatinase-associated lipocalin (NGAL), siderocalin, or 24p3, is a member of the lipocalin superfamily and a pleiotropic mediator of several facets of metabolism [93,94]. LCN2 was recently identified as a bone-derived hormone that regulates appetite through its actions in the mediobasal hypothalamus [95]. Specifically, Mosialou and colleagues demonstrated that peripherally produced LCN2 is able to cross the blood–brain barrier and bind directly to the type 4 melanocortin receptor (MC4R) to induce satiety under physiologic conditions [95]. The literature also demonstrates nutrition-independent metabolic effects of LCN2, as white adipose expression of *lcn2* was demonstrated to activate brown adipose tissue (BAT) via a norepinephrine-independent pathway, as BAT from *lcn2*-KO mice is less thermogenically active [96]. A recent report by Meyers and colleagues expanded upon this notion of Lcn2 activating thermogenic programs using an in vitro approach, demonstrating exogenous Lcn2 increases beiging (*Tbx1* and *Zic1)* and thermogenic markers (*Ucp1* and *Ppar-γ*) in cultured 3T3-L1 adipocyte cells [97]. Since LCN2 is demonstrated to mediate appetite and fat metabolism, two important components of cancer cachexia, our lab sought to address if LCN2 influences any of the signs or symptoms of cancer cachexia.

Utilizing pancreatic ductal adenocarcinoma (PDAC)-associated cancer cachexia models, we demonstrated a clear appetite-regulatory role of LCN2 during the progression of PDAC cachexia [98]. Genetic deletion of *lcn2* improved feeding behaviors and mitigated fat and muscle wasting. However, unlike GDF-15, food-restricting *lcn2*-KO mice diminished these muscle and fat-sparing effects, suggesting LCN2 mediates weight loss and cachexia through its anorectic effects alone, although future studies are still needed. Numerous questions arise from the studies described in this report. How is LCN2 modulating iron trafficking in the context of pancreatic cancer? Is the iron-loaded state of LCN2 important in its appetite-regulating effects? Is peripherally produced LCN2 able to directly modulate metabolism, as described in previous reports, during cachexia progression? Finally, are there permissive factors specific to pancreatic cancer that allow for the anorectic effects of LCN2 during cachexia? Specifically, is CNS inflammation requisite in LCN2 MC4R-dependent effects during cachexia? These questions and more are topics of active investigation and extend beyond the scope of this review.

### 5.3. INSL3

Insulin-like peptide 3 (INSL3) is a member of the insulin/relaxin superfamily and classically described as a Leydig and ovarian theca cell-derived protein and critical regulator of male and female reproductive physiology [99]. However, INSL3 is also shown to regulate normal bone and skeletal muscle physiology, including appropriate bone mineralization [100], as well as promotion of skeletal muscle protein synthesis through the Akt/mTOR/S6 pathway [101]. To our knowledge, there were no reports concerning the role of INSL3 during cancer cachexia until recently.

Using a drosophila eye tumor model, Yeom and colleagues demonstrated that tumor-derived Dilp8, the *Drosophila* homologue of INSL3, induces anorexia through the Lgr3 receptor in the brain (Lgr8 being the putative mammalian orthologue). The authors go on to demonstrate that intracerebroventricular injection of INSL3 alone is capable of inducing appetite suppression in mice, serum INSL3 levels were significantly increased in patients with pancreatic cancer cachexia and serum INSL3 levels were negatively correlated to calorie intake in PDAC patients [102]. Although not shown experimentally, the authors suggest that inhibition of Dilp3/INSL3 did not improve cachexia-associated lean and fat mass wasting despite the observed improvement in food intake [103,104]. Therefore, it appears as though Dilp3/INSL3 signaling in the brain specifically improves feeding behaviors without sparing lean or fat mass wasting during cancer cachexia. The authors also propose a distinct signaling mechanism by which the binding of Dilp3/INSL3 to their cognate receptor in the brain increases expression of a novel anorexigenic hormone NUCB1/NUCB2 in the hypothalamic area, including the supraoptic nucleus, lateral hypothalamic area, arcuate nucleus, paraventricular nucleus and parabrachial nucleus. Given these hypothalamic regions play diverse roles in whole-body metabolism, including regulation of the HPA axis and autonomic signaling, it is plausible that the proposed Dilp3/INSL3–Lgr3/Lgr8 mediates other components of cancer cachexia. Future studies are warranted in order to investigate potential nutrition-independent effects of the Dilp3/INSL3–Lgr3/Lgr8 axis during cancer cachexia.

## 6. Conclusions

Herein, we explored past and present research of CNS-based mechanisms of cancer cachexia, providing a conceptual framework for inflammatory, autonomic and neuroendocrine pathways of energy homeostasis. Although we presented inflammatory, autonomic and neuroendocrine concepts of cachexia individually, it is undeniable that their modulation during health and disease are interdependent. For instance, CRH neurons regulate activity of the HPA axis and simultaneously provide regulatory projections to brainstem and spinal cord neurons that regulate sympathetic outflow [105]. CRH neurons are known to regulate SNS inputs to the heart, adipose tissue, liver and other peripheral tissues, all of which are also sensitive to GC levels established by the HPA axis [69,71,105]. Additionally, hypothalamic IL-1β exposure is sufficient for the activation of the HPA axis, GC production and subsequent muscle catabolism [3]. Therefore, components of the CNS-based inflammatory, autonomic and neuroendocrine pathways activated by cancer progression are mechanistically interdependent and display functional redundancy in regulating peripheral tissue catabolism during cachexia. This instance of redundancy also highlights the CNS ability to amplify biological programs in the periphery, such as muscle catabolism, through multiple efferent effector mechanisms. Therefore, encouraging cachexia therapeutics that target the CNS are likely to modulate several of these CNS processes involved in cancer cachexia (summary of clinical trials targeting or partially targeting the CNS during cachexia in Table 2). Although beyond the scope of this review, it is worth noting the likely involvement of local muscle- and fat-neural networks in the end-organ biology of cancer cachexia. Given the complex interplay between central and peripheral nervous systems, disentangling local and central neural mechanisms of cachexia is a challenging, but richly informative, area of future research.

We end this review with a discussion of three recently identified central mediators of cachexia, including GDF15, LCN2 and INSL3, all peripherally derived molecules that bind to their cognate receptors in the brain to drive cachexia symptoms. While all of these novel mediators of cachexia appear to influence food intake, it is possible that they drive other facets of cachexia through inflammatory, autonomic, or endocrine alterations in the brain as discussed in this review. Indeed, the ability of GDF15 to drive autonomic activation and adipose tissue wasting was only recently described, making it a promising target for alleviating cachexia symptoms through multiple means. However, it remains unclear if LCN2 or INSL3 influence other CNS mechanisms of cachexia beyond nutrition at this time. Future research is needed to determine potential nutrition-independent effects of these promising cancer cachexia targets. Nevertheless, given this recent surge in the identification of CNS pathways and molecules that influence cachexia, therapeutically targeting the CNS represents a promising approach for treating this debilitating wasting syndrome.

## Figures and Tables

**Figure 1 cancers-13-03990-f001:**
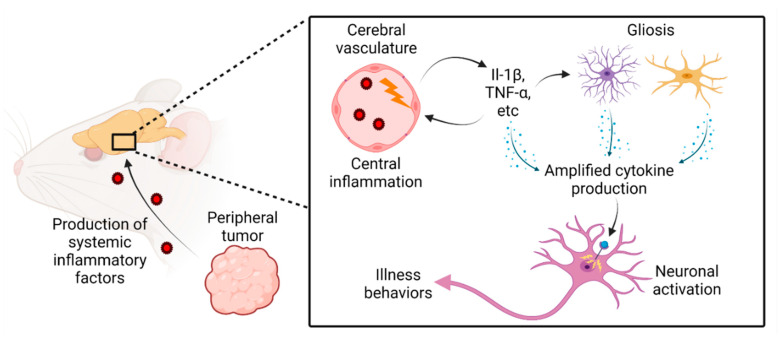
Model of CNS amplification of peripheral inflammatory signals during the evolution of cancer cachexia.

**Figure 2 cancers-13-03990-f002:**
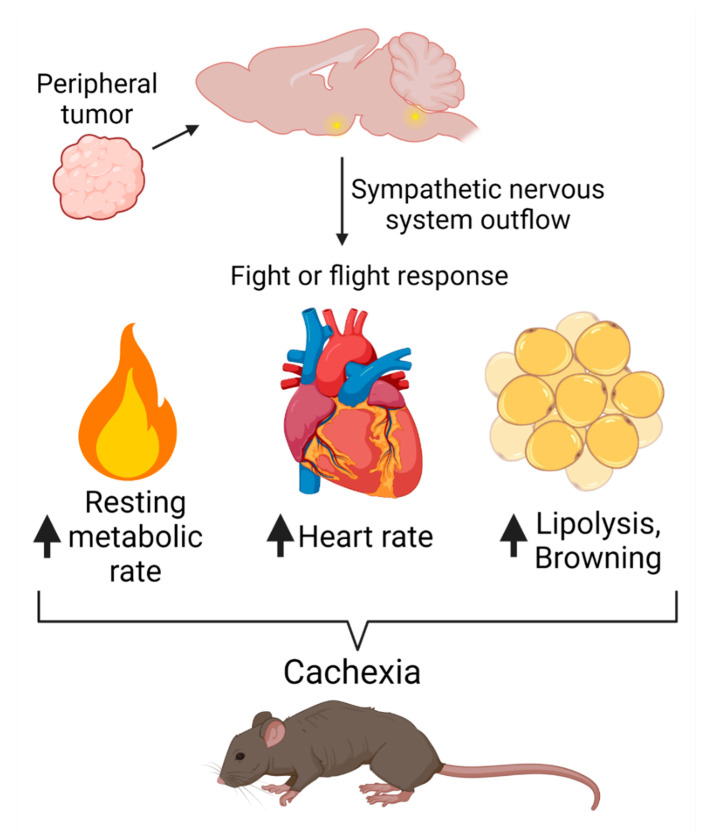
Model and potential mechanisms of sympathetic nervous system engagement in the pathogenesis of cancer.

**Figure 3 cancers-13-03990-f003:**
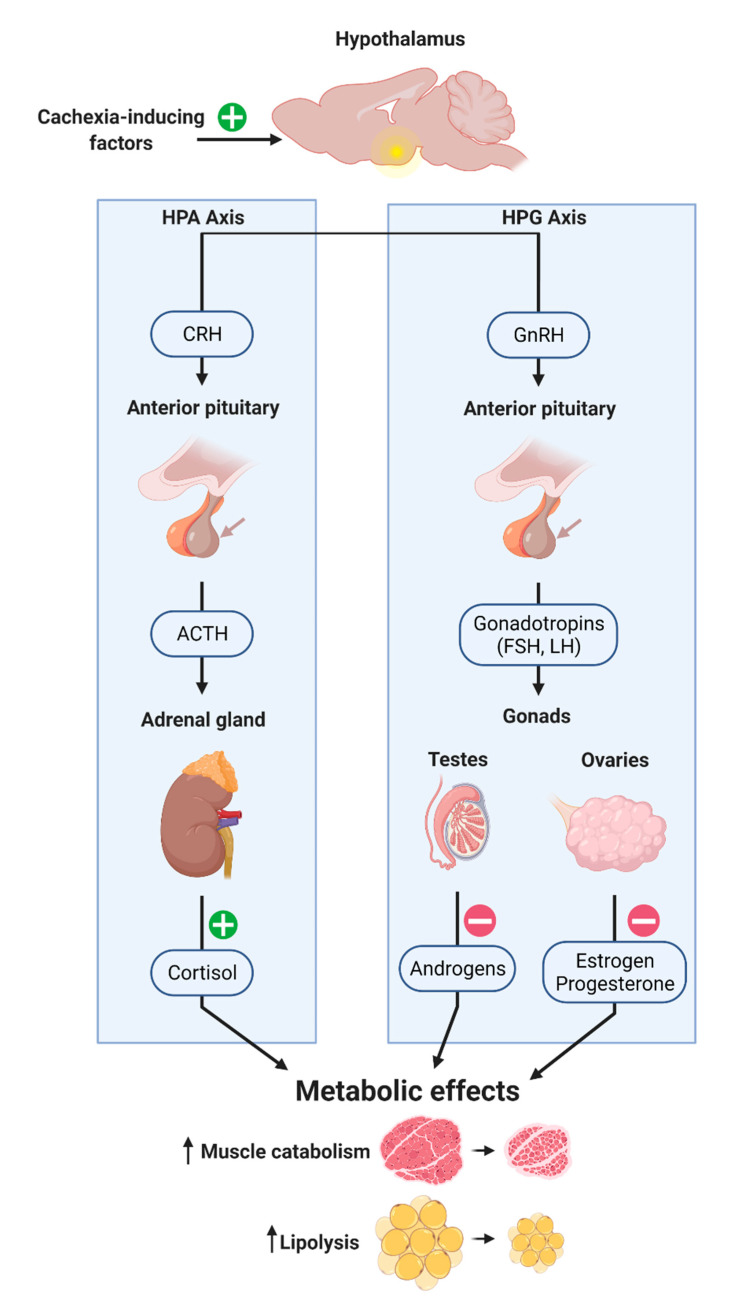
Model and potential mechanisms of hypothalamic–pituitary–adrenal/gonadal axes in the pathogenesis of cancer cachexia.

**Figure 4 cancers-13-03990-f004:**
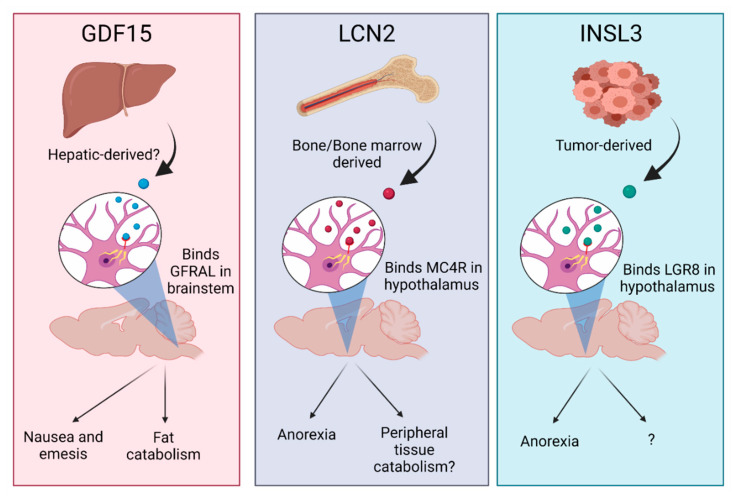
Graphical representation of cachexia-inducing effects of GDF15, LCN2 and INSL3 after binding to their respective receptors in the brain.

**Table 1 cancers-13-03990-t001:** Summary of tumor-induced mediators that interface with the brain to mediate cachexia symptoms.

Tumor-Induced Factor	Effect on CNS Function	References
Il-1b	Modulates neurotransmitter secretion; decreases gluamatergic transmission	[23,24,25]
Il-2	Mediates cognitive decline by hippocampal neurodegeneration, decreases hippocampal acetylcholine secretion and demyelination	[26,27,28]
Il-6	NMDA receptor neurotoxicity; microglial activation	[29,30]
TNF-alpha	Modulates anorexia; increases thermogenesis and respiratory quotient	[31,32,33]
Neutrophils	CNS infiltration via CCR2–CCL2 axis to induce anorexia	[7]
Extracellular vesicles	Axonogenesis, microRNA signaling to induce inflammation	[34,35,36]
Sphingosin-1-phosphate	Promotes anorexia and energy expenditure via persistent activation of hypothalamic STAT3	[37,38]
Serotonin	Inhibits hypothalamic neuropeptide Y secretion	[39]
Macrophage inhibitory cytokine-1	Decreases appetite by interacting with TGF-B type II receptor in the hypothalamus; decreases neuropeptide Y expression and increases POMC expression in arcuate	[40]
Glucagon-like peptide-1	Meditates food intake and body weight by acting on GLP-1R in the brainstem	[41]

**Table 2 cancers-13-03990-t002:** Summary of cachexia clinical trials targeting, in part, the CNS.

CNS-Targeting Drug/Therapy	Mechanism of Action	Clinical Trial Outcomes	References
Anamorelin HCl	Ghrelin receptor agonist	Improved appetite, food intake, body weight and lean mass. No significant improvement in handgrip strength	[106,107]
THC	CB1/CB2 receptors agonist	Improved appetite, food intake, fatigue reversal	[108,109]
Nabilone	CB1/CB2 receptors agonist Synthetic analog of THC	Improved appetite, food intake, decrease in insomnia, pain	[110]
Ghrelin	Orexigenic hormone	Improved appetite, food intake, reduced iatrogenic burden of chemotherapy (nausea, vomiting)	[111]
Megestrol acetate	Progesterone receptor agonist Synthetic analog of progesterone	Improved appetite, food intake, weight gain. Downregulation of proinflammatory cytokines.	[112]
Medroxyprogesterone acetate	Progesterone receptor agonist Synthetic analog of progesterone	Improved appetite, food intake, weight gain. Downregulation of IL-6, IL-1, TNFα	[113]
Celecoxib	COX-2 inhibitor	Weight gain, BMI increase. Downregulation of proinflammatory cytokines. Improvement in handgrip strength, performance	[114,115]
Pentoxifylline/Oxpentifylline	Hemorheological agent/Dimethylxanthine derivative	No significant change in appetite, food intake. Potential negative effects on QoL	[116,117]
Thalidomide	Glutamic acid derivative	Weight gain. Downregulation of proinflammatory cytokines	[118,119,120]
Infliximab	IgG1k monoclonal antibody	No significant change in weight. Potential negative effects on QoL	[121]
Etanercept	TNFα receptor blocker	No significant change in weight	[55]

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
