# Peer review of "Neural Mechanisms of Cancer Cachexia"

_cancers, 2021, doi:10.3390/cancers13163990_

Round 1

Reviewer 1 Report

The entire revision is unsatisfactory. Given this concern, one would anticipate that this paper will have an incermental impact on the field of cancer-driven muscle cachexia

Reviewer 2 Report

Dear Editor,
The authors clearly addressed the reviewer's concerns and written very well. 

Reviewer 3 Report

The authors have very nicely illustrated the subject in the review. Cancer cachexia has debilitating consequence on the body. Although, many reports have identified signaling mechanisms that regulate changes in muscle and adipose tissues, not much focus is given on the neural aspects of cachexia. This review comprehensively discusses both sympathetic nervous system and neuroendrocrine components of cachexia. 

This manuscript is a resubmission of an earlier submission. The following is a list of the peer review reports and author responses from that submission.

Round 1

Reviewer 1 Report

This review papers aims at providing a comprehensive overview of how the central nervous system contributes to cancer cachexia, focusing on the integration and interpretation of inflammatory signals and other pro-cachectic inputs. As such, this paper is well presented and discuss the most relevant literature as it relates to the neural mechanisms of cancer cachexia. Despite these significant strengths, there are many issues that need to be addressed in order to provide the readers with a cohesive understanding of the general mechanisms leading to cancer cachexia.

1) The authors consistently refer to anorexia as an important aspect of cancer cachexia. However, it is well established that cancer cachexia is a catabolic disease that leads to muscle and fat wasting even under conditions where foot intake is increased. With that being said, the authors might need to provide clear definitions of cancer cachexia and anorexia throughout the manuscript. A glossary might help address this issue.

2) The mechanisms leading to muscle wasting and loss of fat mass (including fat browning) are barely discussed in the manuscript. It would be very important to add a concise section about such mechanisms and then discuss how the central nervous system could contribute to those mechanisms irrespective of whether they are direct or indirect.

3) The mechanisms by which tumors impact the central nervous system are also barely discussed. Here again, the authors need to provide a concise description of the factors that allow the tumor to communicate with the brain and how the brain interprets those signals to culminate in cancer cachexia, including within muscle and fat.   

4) A whole section in this manuscript is focused on IL-1b. However, extensive genetic studies have shown that this cytokine plays minimal role in cancer cachexia. Perhaps, this section should be significantly reduced or eliminated and use the space to address the issues pointed in 2 and 3.

5) The authors extensively discussed GDF15, LCN2, and INSL3. However, LCN2 and INSL3 are well known to regulate food intake. Perhaps it would be consistent to discuss mainly their pro-cachectic activities within fat and muscle as well as the underlying mechanisms bridging the central nervous system to those peripheral systems.

6) It would be more informative to discuss the possible contribution of local muscle-neural and fat-neural networks to cancer cachexia.

7) The link between metastasis (including in brain) and cancer cachexia is not discussed.

8) There is some redundancy between the abstract and the introduction.

Reviewer 2 Report

The manuscript by Brennan Olson et al., entitled “Neural Mechanisms of Cancer Cachexia,” was written very well and scientifically sounds excellent. They summarize the cellular and molecular mechanisms of CNS dysfunction during cancer cachexia, focusing on inflammatory, autonomic, and neuroendocrine processes and ending with a discussion of recently identified CNS mediators of cachexia, including GDF15 LCN2 and INSL3. This is a fascinating review article and suitable for cancers journal. However, I have a minor concern that needs to address.

  1. The author should provide or discuss any therapeutic intervention for CNS-induced cachexia.